

# Chemically mediated rheotaxis of endangered tri-spine horseshoe crab: potential dispersing mechanism to vegetated nursery habitats along the coast

Kit Yue Kwan[1,*], Xin Yang[1,*], Chun-Chieh Wang[2], Yang Kuang[1], Yulong Wen[1], Kian Ann Tan[1], Peng Xu[1], Wenquan Zhen[1], Xueping Wang[1], Junhua Zhu[1] and Xing Huang[1]

[1] College of Marine Sciences, Beibu Gulf Ocean Development Research Centre, Guangxi Key Laboratory of Beibu Gulf Biodiversity Conservation, Beibu Gulf University, Qinzhou, Guangxi, China
[2] Guangxi Key Laboratory of Marine Environmental Science, Guangxi Beibu Gulf Marine Research Center, Guangxi Academy of Sciences, Nanning, China
[*] These authors contributed equally to this work.

Corresponding author
Chun-Chieh Wang,
chunchiehwang@gxas.cn

## ABSTRACT

**Background**. An enhanced understanding of larval ecology is fundamental to improve the management of locally depleted horseshoe crab populations in Asia. Recent studies in the northern Beibu Gulf, China demonstrated that nesting sites of Asian horseshoe crabs are typically close to their nursery beaches with high-density juveniles distributed around mangrove, seagrass and other structured habitats.

**Methods**. A laboratory Y-maze chamber was used to test whether the dispersal of early-stage juvenile tri-spine horseshoe crab *Tachypleus tridentatus* is facilitated by chemical cues to approach suitable nursery habitats. The juvenile orientation to either side of the chamber containing controlled seawater or another with various vegetation cues, as well as their movement time, the largest distance and displacement were recorded.

**Results**. The juveniles preferred to orient toward seagrass *Halophila beccarii* cues when the concentration reached $0.5 \text{ g l}^{-1}$, but ceased at $2 \text{ g l}^{-1}$. The results can be interpreted as a shelter-seeking process to get closer to the preferred settlement habitats. However, the juveniles exhibited avoidance behaviors in the presence of mangrove *Avicennia marina* and invasive saltmarsh cordgrass *Spartina alterniflora* at $2 \text{ g l}^{-1}$. The juveniles also spent less time moving in the presence of the *A. marina* cue, as well as reduced displacement in water containing the *S. alterniflora* cue at 1 and $2 \text{ g l}^{-1}$. These results may explain the absence of juvenile *T. tridentatus* within densely vegetated areas, which have generally higher organic matter and hydrogen sulfide.

**Conclusion**. Early-stage juvenile *T. tridentatus* are capable of detecting and responding to habitat chemical cues, which can help guide them to high-quality settlement habitats. Preserving and restoring seagrass beds in the intertidal areas should be prioritized when formulating habitat conservation and management initiatives for the declining horseshoe crab populations.

## INTRODUCTION

Horseshoe crabs are an ancient group of invertebrates that are broadly distributed along the west coast of the North Atlantic and Pacific Oceans. They are inshore species which are important in the food web of coastal and estuarine ecosystems (*Botton, 2009*). Their eggs serve as protein and lipid sources for fishes and migratory shorebirds (*Mizrahi & Peters, 2009*), whereas the juveniles and adults are key predators of the benthic community in intertidal flats (*Gaines et al., 2002*; *John et al., 2012*; *Kwan et al., 2021*). However, horseshoe crabs are heavily harvested for their blood for the manufacture of *Tachypleus* and *Limulus* amebocyte lysates, the worldwide standardized tests for bacterial endotoxin detection in pharmaceutical products (*Gauvry, 2015*; *Tinker-Kulberg et al., 2020*). In addition to resource exploitation, habitat loss and degradation from coastal development (*Tsuchiya, 2009*; *Nelson et al., 2015*; *Wang et al., 2020*), as well as bycatch by artisanal fishing and discarded fishing gear, are also widely observed to cause considerable threats to horseshoe crab populations (*Zauki et al., 2019*; *Wang et al., 2022*). The Atlantic horseshoe crab *Limulus polyphemus* and tri-spine horseshoe crab *Tachypleus tridentatus* are listed as "Vulnerable" (*Smith et al., 2016*) and "Endangered" (*Laurie et al., 2019*), respectively, in the IUCN Red List of Threatened Species, while the status of other two Asian species, the coastal horseshoe crab *T. gigas* and mangrove horseshoe crab *Carcinoscorpius rotundicauda* are under reassessment owing to the recent reports describing substantial population declines (*John et al., 2018*; *Wang et al., 2020*). To reverse the declining trend, national and regional conservation measures have been imposed in Bangladesh, India, China, Singapore, Indonesia, and in specific regions in Japan. The effectiveness of these measures in protecting the remaining horseshoe crab populations may be limited (*Wang et al., 2020*), possibly due to insufficient scientific knowledge, financial resources and enforcement capacity (*Xie et al., 2020*).

The larval dispersal and settlement of marine species are critical for the persistence of local populations; therefore, an enhanced understanding of their ecology and behavior, particularly for endangered or locally depleted species, is useful for management and conservation (*Botton & Loveland, 2003*; *Green et al., 2015*; *Whomersley et al., 2018*). Horseshoe crabs have unique reproductive strategies to maximize egg hatching success and subsequent larval development (*Penn & Brockmann, 1994*; *Vasquez et al., 2015*). The spawning pairs in amplexus migrate from shallow waters to sandy estuarine beaches, and lay clusters of eggs beneath the sediment in the intertidal zones (*Smith et al., 2017*). The eggs hatch into planktonic trilobite larvae and settle in the vicinity of the shoreline (*Botton & Loveland, 2003*; *Botton, Tankersley & Loveland, 2010*). Most hatched larvae emerge from the sediment at high spring tides when the water reaches the height of the nests (*Botton & Loveland, 2003*; *Ehlinger, Tankersley & Bush, 2003*), facilitating larval dispersal from the nesting locations.

While the spawning biology of horseshoe crabs may share common characteristics, the existing information for Asian species is limited and mostly descriptive. Similar to their Atlantic counterpart, the distribution of newly settled and early-stage juvenile *T. tridentatus* and *C. rotundicauda* populations is non-random and has a high tendency to

stay close to mangrove, seagrass and other structured habitats (*Kwan et al., 2016*; *Kaiser & Schoppe, 2018*; *Xie et al., 2020*; *Meilana, Hakim & Fang, 2021*). Recent spawning habitat surveys in the northern Beibu Gulf, following the last report in 1984 in China (*Cai, Lin & Huang, 1984*), demonstrated that the identified nesting beaches were adjacent to nursery habitats for juveniles (*Kwan et al., 2022*). Little is known regarding the movement behavior of the larvae and early-stage juveniles under field conditions. Previous laboratory studies on *L. polyphemus* suggest that their directed movements to water flow (*i.e.,* rheotaxis) change upon exposure to habitat chemical cues (*Medina & Tankersley, 2010*; *Butler & Tankersley, 2020*). A rheotaxis can either be positive by turning face into the current to hold their position rather than being swept downstream, or negative to avoid oncoming currents (*Kobayashi et al., 2014*). In the experiment of *Butler & Tankersley (2020)*, *L. polyphemus* larvae exhibited a positive rheotaxis in the presence of chemical cues from seagrass associated with their settlement sites, which may imply that the strong tendency of early juveniles to remain close to the beach is a consequence of upstream movement behavior mediated by habitat chemical cues. However, the mechanism of post-larval orientation and settlement is likely species- and/or site-specific, depending on the perceived coastal environmental conditions (*Rossi et al., 2019a*).

In this study, we examined whether the early-stage juvenile *T. tridentatus* are able to detect and respond to chemical cues associated with varied coastal vegetations available in their nursery habitats. We predict that the habitat chemical cues can influence the orientation and movement behaviors of juvenile, providing guidance to preferred settlement habitats, which shapes the distribution patterns of early juveniles in the immediate vicinity of the shoreline. The Beibu Gulf, a semi-closed gulf located off the coast of southern China and northern Vietnam, is broadly considered to be one of the most important habitats for the remaining high-density population of endangered *T. tridentatus* (*Brockmann & Smith, 2009*; *Sekiguchi & Shuster, 2009*; *Liao et al., 2019*). The spawning and nursery habitats of Asian horseshoe crabs in the gulf are typically characterized by extensive mangrove fringes along the coastline with patches of seagrass *Halophila* spp. and invasive saltmarsh cordgrass *Spartina alterniflora* scattered on the intertidal flats (*Xie et al., 2020*; *Kwan et al., 2022*). These characteristics of the spawning and nursery habitats serve as a good opportunity to test our prediction of the orientation and movement behaviors of the endangered *T. tridentatus* juveniles for exploring the ecological importance to settlement in suitable habitats.

## MATERIALS & METHODS

### Larval and juvenile horseshoe crab rearing

*Tachypleus tridentatus* larvae were obtained from the Guangxi Institute of Oceanology, China. The use of hatchery-bred animals was approved by the Department of Agriculture and Rural Affairs of Guangxi Region, China (approval number 2022-0131). Mating pairs of *T. tridentatus* were kept in indoor tanks with an approximately 10-cm sediment layer underneath. The released eggs were incubated in hanging baskets from the surface of culture water with continuous, vigorous airflow pumping below the baskets (*Xu et al.,*

*2021*). Most eggs developed and hatched into trilobite larvae after one-month rearing under the following environmental conditions: temperature 26–30 °C, salinity 32–33 ppt, pH 7.6–7.9, dissolved oxygen 6–7 mg l$^{-1}$.

The hatched larvae were transported to the laboratory and cultured in aquarium tanks (dimension: 120 × 40 × 25 cm) equipped with a water filtration system, thermostatic heaters and ultraviolet sterilizers. A 4-cm sediment layer was provided underneath. Seawater was maintained at the rearing conditions similar to egg incubation. The water quality was monitored weekly, and half of the volume of water was changed every month or whenever water ammonia concentration was above 0.1 mg l$^{-1}$. Frozen brine shrimp larvae were provided thrice per week when the larvae had developed into second-instar juveniles.

## Experimental setup and conditioned water preparation

The experimental setup consisted of a laboratory Y-maze acrylic chamber and two reservoirs containing control and conditioned waters, separately (Fig. 1A). A water pump was placed within each reservoir to pump the test waters into the inflow end at each side of the Y-maze chamber. The chamber was filled with seawater to six cm depth with a 1-cm sand layer underneath, so as to keep all experimental juveniles completely submerged under the water. Prior to the experiment, two acrylic movable plates were inserted near the outflow end of the chamber (Fig. 1A) to maintain the water level and avoid the immediate mixing between control and conditioned waters. The experiment began after the experimental waters had been flowing in the chamber for at least 10 mins. The flow rate was calculated by measuring the volume of outflowing seawater per unit time. A standard flow rate (200 mL/min) was maintained throughout the experiments by adjusting the control valve on each water tube connected to the water pumps until reaching stable equilibrium from each side with the aid of different dyed waters (water-soluble ink). Two video cameras were installed on each side to record juvenile directional movements relative to the flow of water.

Conditioned waters were prepared using three dominant vegetation sources, including mangrove *Avicennia marina*, seagrass *Halophila beccarii* and saltmarsh cordgrass *Spartina alterniflora*, which can be found in *T. tridentatus* nursery habitats along the coast of the northern Beibu Gulf, China (*Xie et al., 2020*). Fresh fallen leaves of mangrove, seagrass and saltmarsh cordgrass were collected at the identified nursery sites (*Kwan et al., 2021*) during low tides in the summer (May–September) of 2020. The collected samples were rinsed repeatedly, freeze-dried for at least one week, and ground into the powder with a mortar and pestle. The dried samples were weighed, dissolved into artificial seawater at salinity 30 ppt, homogenized and filtered after 12 h, to prepare the conditioned waters at concentrations of 0.25, 0.50, 1.00 and 2.00 g l$^{-1}$. The levels were selected based on the concentration range (0.3–30 g l$^{-1}$) described in *Butler & Tankersley (2020)*). However, the preparation method of conditioned seawater in the present study (dissolution of ground vegetation powder) was slightly different from those in the previous study (24-h incubation of fresh vegetation), which should cause different actual levels of chemical cues available in the conditioned waters. The trials with concentrations higher than 2 g l$^{-1}$ were not conducted because the conditioned water would become too turbid and the juvenile

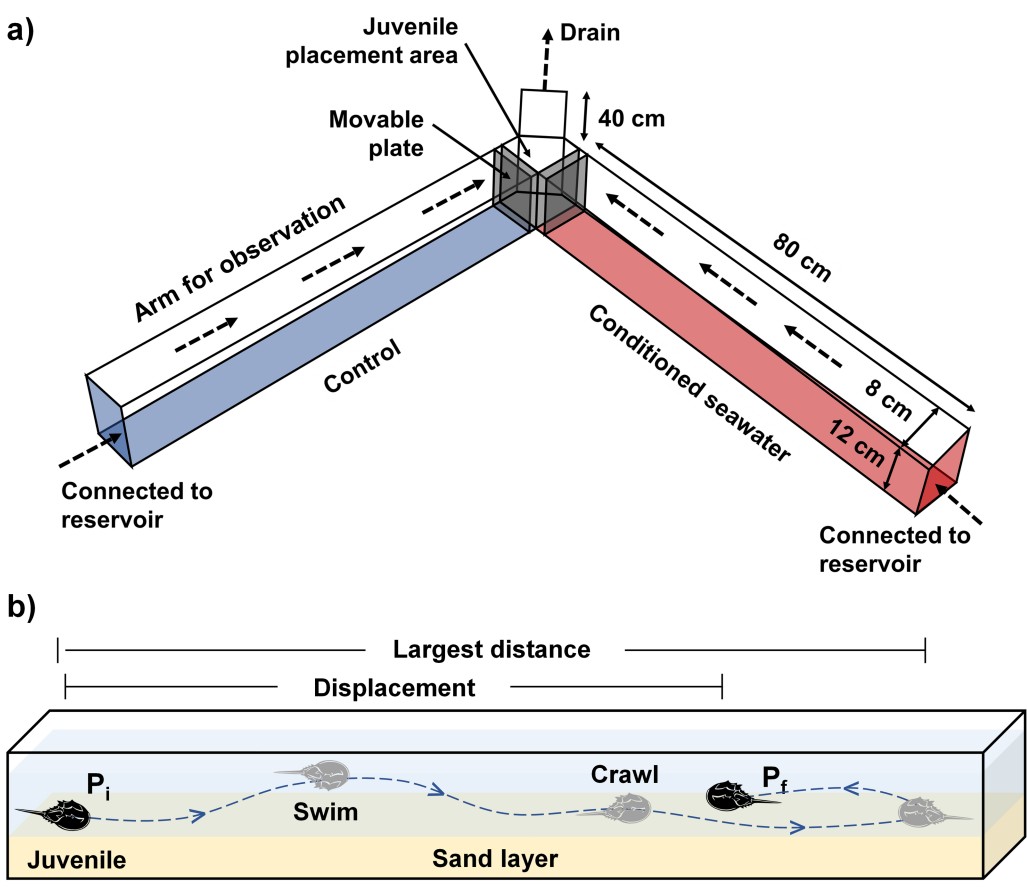

**Figure 1** (A) **The experimental setup comprises a Y-maze chamber used to measure the rheotaxis and movement behavior of juvenile *T. tridentatus* in response to control (blue) *versus* conditioned seawater with chemical cues (red).** (B) **A typical example of juvenile movement in the chamber.** Direction of water flow is indicated by dashed arrows. The movement pathway is indicated by a dashed line. $P_i$ = initial position, $P_f$ = final position when the allowed experimental time ended, *i.e.*, 30 mins.

behavioral parameters could not be quantified. Artificial seawater at salinity 30 ppt was used as the control. All experimental waters were subjected to experiments within 12 h of preparation.

## Orientation and movement behaviors toward chemical cues

To quantify the movement responses during the settlement process, the orientation and various behavioral data from 60 juveniles were collected per treatment. During each treatment, a second-instar juvenile *T. tridentatus* (prosomal width: 7.5–8.8 mm, wet weight: 35.9–55.3 mg) was randomly chosen and introduced into the intersection area of the chamber (Fig. 1A). The juvenile was given 30 mins to respond to the flow by moving upstream to either side of the chamber containing control or conditioned water, or downstream toward the outflow end. The orientation, movement time, the largest movement distance and displacement of the juvenile were quantified based on the video recordings. After the completion of data collection from 10 juveniles, the inner surface

and sand layer of the chamber was rinsed completely. Another group of 10 juveniles was used for the same treatment by alternating the inflow of conditioned water from the left to the right arm of the chamber, to test if the choice of seawater source by the juveniles was non-random. The set of experiment was repeated three times, and all juveniles were only used once per observation (each treatment: 10 juveniles ×2 positions ×3 replicates).

Because none of the experimental juveniles traveled downstream throughout the experiment, the orientation parameter was used to quantify the percentage of individuals moving upstream to choose control/conditioned water. The orientation toward conditioned or control water of each juvenile was recorded by a single video, and the percentage of individuals moving up to either side of the Y-maze chamber was calculated based on the video recordings from 10 different juveniles. A juvenile that failed to travel in either direction during the first 10 mins was considered "unresponsive" and would be replaced by another juvenile. The proportion of "unresponsive" juveniles was very low, which ranged from 0–1 individuals in each experimental replicate. Movement time was the total time the juveniles spent crawling or swimming in the chamber. The largest movement distance was defined as the longest length traveled in a single upstream direction, whereas displacement was the length between the initial and final points of movement within the allowed experimental time, *i.e.*, 30 mins (Fig. 1B). Artificial lighting was used to ensure that all animals were exposed to the same conditions. None of the juveniles were sacrificed during the experiment, and the study protocol was approved by the Committee for Animal Welfare of the Beibu Gulf University.

## Statistical analysis

Data were first examined for normality and homogeneity of variance by Shapiro–Wilk and Levene's tests, respectively. Student's *t* and Mann–Whitney U tests were conducted to check whether the choice of the left/right arm of the chamber by the juveniles was non-random. The data from two groups of 10 juveniles were pooled for subsequent behavioral parameter analyses after the differences were found to be statistically similar (Table S1). Since the orientation data were non-normal, non-parametric binomial tests were performed to examine the possible differences in juvenile orientation between control and conditioned waters at individual concentration. The test proportion of the binomial model was set at 0.50. Student's *t* tests were used for other behavioral parameters analyses. To understand the overall effects of various vegetation sources at different concentrations on juvenile behaviors, the data were analyzed using two-way analysis of variance (ANOVA: source [fixed] × concentration [fixed]). Multiple pair-wise comparisons among sources/concentrations were applied using *post hoc* Tukey's tests with Bonferroni adjustments when a significant difference was identified. All the above analyses were implemented using IBM SPSS Statistics Software (version 26; Armonk, NY, USA).

## RESULTS

Between 17%–82% of juvenile *T. tridentatus* traveled to the side containing habitat cues from different vegetation sources at various concentrations (Figs. 2A–2C). Binomial tests between control and treatment groups revealed that statistically higher proportions of

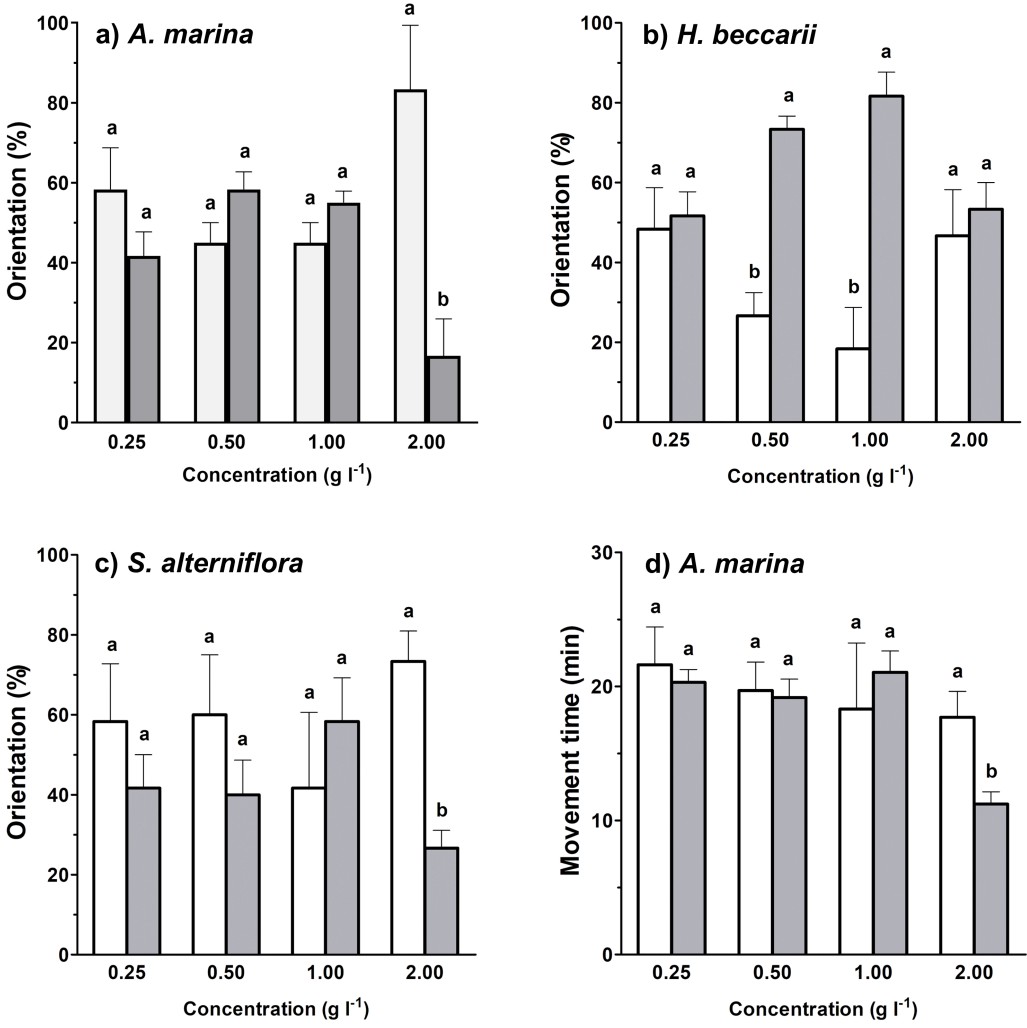

**Figure 2  Effects of each vegetation source at varying concentrations compared to the control on rheotaxis (A–C), and movement time (D) of juvenile *T. tridentatus*.** Movement time of the juveniles in seagrass *H. beccarii* and saltmarsh cordgrass *S. alterniflora* cues was statistically similar to those in the control, and was not shown in the figure. The data are expressed in mean ±standard deviation. Different lowercase letters represent statistical differences ($p < 0.05$) between control (white bars) and conditioned (grey bars) waters at the corresponding concentration.

juveniles responded to *H. beccarii* chemical cues at 0.50 and 1.00 g l$^{-1}$, while significantly lower percentages of juveniles moved upstream approaching *A. marina* chemical cues at 2.00 g l$^{-1}$ and *S. alterniflora* chemical cues at 2.00 g l$^{-1}$ (Figs. 2A–2C). For other behavioral parameters, a significant reduction in juvenile movement time for >57% was detected in water containing *A. marina* cue at 2.00 g l$^{-1}$, while the juvenile largest movement distance and displacement performed in seawater sources with chemical cues were similar to those recorded in the control (Table 1, Fig. 2D).

When the overall effects of various vegetation sources at different concentrations were simultaneously considered, both source and concentration of chemical cues were found

**Table 1 Statistical results of Student's *t* tests ($d.f. = 4$) showing the effects of each vegetation source at varying concentrations compared to the control on various movement behaviors of juvenile *T. tridentatus*.**

| Habitat cue | Conc. (g l$^{-1}$) | Movement time | | Largest distance | | Displacement | |
|---|---|---|---|---|---|---|---|
| | | *t* | *p* | *t* | *p* | *t* | *p* |
| *A. marina* | 0.25 | −0.825 | 0.413 | 0.196 | 0.846 | 0.248 | 0.805 |
| | 0.50 | −0.169 | 0.866 | −1.725 | 0.090 | −0.935 | 0.354 |
| | 1.00 | 1.446 | 0.153 | 0.672 | 0.504 | 1.668 | 0.101 |
| | 2.00 | −3.081 | **0.003** | −1.033 | 0.306 | −1.314 | 0.194 |
| *H. beccarii* | 0.25 | 0.088 | 0.930 | 0.987 | 0.328 | 0.474 | 0.637 |
| | 0.50 | −0.857 | 0.395 | −0.283 | 0.778 | −0.568 | 0.572 |
| | 1.00 | 1.175 | 0.245 | 0.450 | 0.655 | −0.015 | 0.988 |
| | 2.00 | 1.018 | 0.313 | 0.335 | 0.739 | 0.383 | 0.703 |
| *S. alterniflora* | 0.25 | −0.712 | 0.479 | 1.507 | 0.137 | 0.595 | 0.554 |
| | 0.50 | 0.117 | 0.907 | 0.118 | 0.907 | 0.313 | 0.755 |
| | 1.00 | −1.71 | 0.093 | −0.571 | 0.570 | −0.227 | 0.821 |
| | 2.00 | −0.279 | 0.781 | −1.254 | 0.215 | −1.225 | 0.226 |

**Notes.**
Significant *p* values (<0.05) are highlighted in bold.
Conc., Concentration.

to significantly alter the displacement of juveniles, but only source and concentration were noted to affect juvenile movement time and the largest distance, respectively (Table 2, Fig. 3). A significant decrease in juvenile displacement at 1.00 and 2.00 g l$^{-1}$ was also observed in water containing *S. alterniflora* cue (Fig. 3A). In terms of movement time, the juveniles were more active in seawater containing *H. beccarii* cue than those in *A. marina* (Fig. 3B). A significant reduction in the largest movement distance of juveniles was also recorded at 2 g l$^{-1}$, compared to those at 0.25 g l$^{-1}$, regardless of the vegetation sources (Fig. 3C).

## DISCUSSION

There is increasing interest in studying horseshoe crab populations due to their biomedical importance and use in various fisheries, and understanding the factors that may contribute to larval recruitment is a worthwhile investigation. Recent studies provided useful information on the nesting/nursery habitat distributions and larval hatching processes of Asian horseshoe crabs (*Itaya et al., 2022*; *Kuang et al., 2022*; *Kwan et al., 2022*). However, little is known regarding the role of chemoreceptive and olfactory capabilities in larval transport and settlement, despite the fact that high densities of juvenile Asian horseshoe crabs are known to occur in the upper intertidal beaches adjacent to mangrove, seagrass and other structured habitats (*Xie et al., 2020*). In this study, the use of chemical cues in seeking preferred settlement habitat by *T. tridentatus* was tested using a laboratory Y-maze chamber. Our results provided evidence that early-stage juvenile *T. tridentatus* are capable of detecting and responding to chemical cues associated with the typical vegetations available in nursery habitats. Overall, the juveniles were attracted to the seagrass *H. beccarii* cue when the concentration reached 0.5 or 1 g l$^{-1}$. On the other hand, the juveniles tended

**Table 2** Statistical results of two-way ANOVA showing the effects of different source and concentration of chemical cues (g l⁻¹) on movement behaviors of juvenile *T. tridentatus*.

| Response | d.f. | F | p |
|---|---|---|---|
| **Movement time (min)** | | | |
| Source | 2 | 4.344 | **0.025** |
| Conc. | 3 | 0.751 | 0.533 |
| Source × Conc. | 6 | 8.316 | **<0.001** |
| Error | 24 | | |
| **Largest distance (cm)** | | | |
| Source | 2 | 2.488 | 0.104 |
| Conc. | 3 | 3.090 | **0.046** |
| Source × Conc. | 6 | 1.756 | 0.151 |
| Error | 24 | | |
| **Displacement (cm)** | | | |
| Source | 2 | 4.728 | **0.019** |
| Conc. | 3 | 4.884 | **0.009** |
| Source × Conc. | 6 | 2.398 | 0.060 |
| Error | 24 | | |

**Notes.**
Significant *p* values (<0.05) are highlighted in bold.
d.f., Degree of freedom.

to avoid chemical cues from mangrove *A. marina* and saltmarsh cordgrass *S. alterniflora* at relatively high concentrations (*i.e.*, 2 g l⁻¹ in this study).

The use of chemical cues was documented in examples of marine decapod crustaceans and fish (*Havel & Fuiman, 2016*; *Foretich et al., 2017*; *Hinojosa et al., 2018*; *Arvedlund & Kavanagh, 2009*). Horseshoe crabs are known to possess a variety of chemoreceptors on the gills, flabellum, chilaria and walking legs, which would respond to oxygen in seawater and varying chemical cues associated with food (*Quinn, Paradise & Atema, 1998*; *Mittmann & Scholtz, 2001*; *Saunders et al., 2010*). In Cape Cod, U.S.A., *L. polyphemus* were observed to locate their preferred food, *Mya arenaria*, which were completely buried within the sediment (*Smith, 1953*). There is also evidence of chemical cue use by male *L. polyphemus* in locating spawning females. *Hassler & Brockmann (2001)* found that a cement model with conditioned seawater collected from spawning females was more attractive to males. Previous studies also demonstrated that *L. polyphemus* would use other sensory cues in addition to chemical cues to adapt to the overall complexity of signals in coastal and estuarine environments. The use of visual cues enables male *L. polyphemus* to see and respond to females at night (*Barlow, Ireland & Kass, 1982*; *Herzog, Powers & Barlow, 1996*), and pair up with larger females (*Hassler & Brockmann, 2001*; *Barlow & Powers, 2003*).

Relatively little is known about the use of multisensory cues by larvae and juvenile horseshoe crabs to identify preferred habitats. *Limulus polyphemus* larvae were noted to be more active at nighttime and positively phototactic to dim light sources such as moonlight (*Rudloe, 1979*; *Botton & Loveland, 2003*). The major releases of hatched larvae from the nesting sites are shown to be associated with high water conditions such as hydration, hypoosmotic shock and agitation (*Ehlinger & Tankersley, 2003*; *Botton, Tankersley &*

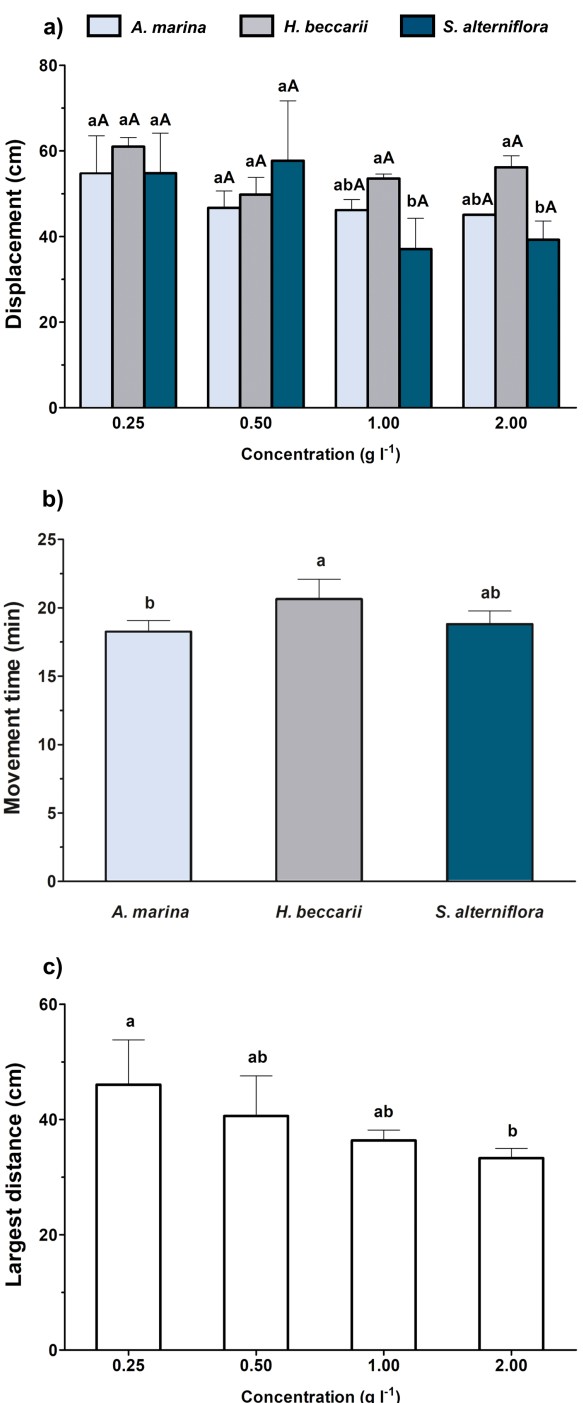

**Figure 3 Movement behaviors (mean ± standard deviation) of juvenile *T. tridentatus* exposed to varying habitat chemical cues.** The data were tested by two-way ANOVA, followed by multiple post hoc Tukey tests with Bonferroni adjustments. Different lowercase letters represent statistical differences ($p < 0.05$) among vegetation sources at the same concentration, whereas different capital letters indicate significant differences among concentrations of individual source (A). Since only source and concentration were noted to affect juvenile movement time (B) and the largest distance (C), respectively, the statistical differences among treatment groups are indicated by different lowercase letters.

*Loveland, 2010*; *Kuang et al., 2022*). These exogenous cues are possibly detected by mechanoreceptors available on the entire surface of prosoma, spines and walking legs (*Wyse, 1971*), to facilitate the dispersal of larvae away from the spawning locations. Our results, together with previous studies on *L. polyphemus* (*Medina & Tankersley, 2010*; *Butler & Tankersley, 2020*), suggest that chemical cues are involved in the settlement and habitat selection process. Horseshoe crab larvae and juveniles were more directed toward chemical cues from seagrasses (*Medina & Tankersley, 2010*; *Butler & Tankersley, 2020*). The responses were perceived as a shelter-seeking behavior, since high-density juvenile *T. tridentatus* populations in the northern Beibu Gulf, China were found in areas of seagrass patches, mainly *Halophila* species (*Xie et al., 2020*). Apart from providing refuge from predation, other studies also revealed that the juveniles predominantly assimilated energy from seagrass as basal production sources in the food web (*Kwan, Cheung & Shin, 2015*; *Fan et al., 2017*; *Kwan et al., 2021*). However, it is rare to find juvenile Asian horseshoe crabs near *S. alterniflora* in the field, even though the invasive plant has expanded rapidly throughout the Chinese coastline (*Meng et al., 2020*), and highly overlapped with horseshoe crab habitats (*Kwan, Cheung & Shin, 2015*; *Xie et al., 2020*; *Kwan et al., 2021*). Our data also showed that juvenile *T. tridentatus* showed stronger preferences for the native *H. beccarii* habitat over the one with invasive *S. alterniflora*.

Although not addressed in our study, biofilm available on the plants and other substrata can also act as settlement cues for a broad variety of marine invertebrate larvae, including mollusks (*Liang et al., 2020*), crustaceans (*Siddik & Satheesh, 2019*), polychaetes (*Freckelton et al., 2022*), gastropods (*La Marca et al., 2018*), cnidarians (*Petersen et al., 2021*) and echinoderms (*Huggett et al., 2006*). Marine biofilms are complex, heterogenic microbial communities, mainly bacteria and diatoms, surrounded by a matrix of extracellular polymeric substances (*Antunes, Leão & Vasconcelos, 2019*). Larval settlement responses to different bacteria can be species-specific. Similarly, the bacterial community on the surfaces and/or roots of habitat plants can also be important as settlement and behavioral cues for early-stage *T. tridentatus*. While the role of bacteria in larval settlement of horseshoe crabs is currently unclear, their larvae and early-stage juveniles are known to feed primarily on sedimentary organic matter (*Gaines et al., 2002*; *Kwan et al., 2021*), which is dominated by benthic diatoms (*e.g.*, Naviculaceae and Cymbellaceae in Beibu Gulf region, Table S2). Alternatively, chemical compounds released during the decay processes of coastal plants may also attract the settlement of marine invertebrate larvae, as seen in several amphipod species associated with seagrass bed (*Edgar, 1992*) and the mangrove jellyfish *Cassiopea xamachana* (*Hofmann, Fitt & Fleck, 1996*; *Fleck & Fitt, 1999*). However, the preparation of conditioned seawater using dried plant materials in the current study may lower the effects of live bacteria and decayed plant compounds on settlement behaviors of the early-stage juvenile *T. tridentatus*.

In this study, the chemically-mediated orientation and movement behaviors of the juveniles were generally concentration-dependent. As noted in Fig. 2B, the juveniles preferred the water containing seagrass chemical cues at a concentration of 0.5 and 1 g l$^{-1}$. However, the effect on directional choice toward seagrass cue was ceased at 2 g l$^{-1}$ and became statistically similar to that observed in control water. The results can be

interpreted as the movement process of the juveniles seeking settlement habitats (*Medina & Tankersley, 2010*): their movements become more directed when the juveniles get closer to the source, which is indicated by the increased concentration of seagrass chemical cues. When the concentration is too high (*e.g.*, 2 g l$^{-1}$ in this study), the juveniles may perceive the signal as the arrival to the preferred settlement habitats, and therefore their behavioral responses would become weaker. In contrast, the juveniles exhibited avoidance behaviors when getting too close to mangrove and saltmarsh cordgrass, as indicated by selecting the side with control seawater when the source concentrations had reached 2 g l$^{-1}$. Other movement behaviors, including reduced time spent on movement and/or shorter displacement, also recorded a similar trend (Figs. 2D and 3A). A possible explanation for such avoidance behaviors is that the densely vegetated saltmarsh cordgrass and mangrove areas have slower water movement and accumulation of the fine-grained, poorly drained substratum, which would result in higher concentrations of organic matter and hydrogen sulfide in the areas (*Wang et al., 2015*; *Rossi et al., 2019b*; *Su et al., 2020*; *Li et al., 2021*). A recent study in the Beibu Gulf region also demonstrated that the *Spartina* occupation reduced the diversity of macroinvertebrate assemblages on intertidal flats (*Su et al., 2020*), and therefore may affect the availability of food sources for the juveniles (*Kwan et al., 2021*). The presence of high tannin, phenolics and other plant defensive compounds in *A. marina* and *S. alterniflora* extracts (*Zhou et al., 2010*; *Zhang et al., 2021*) were found to negatively affect benthic invertebrates (*Alongi, 1987*; *Lee, 1999*), probably also reducing the rheotaxis of juvenile *T. tridentatus* toward these vegetations.

While the induction by a single source of vegetation cues never exists in the marine environment, and the actual contribution of these factors in the field is poorly understood, similar laboratory experiments are common and useful to investigate the mechanism of larval settlement and habitat selection in marine invertebrates (*Suárez-Rodríguez, Kruesi & Alcaraz, 2019*; *Gravinese et al., 2020*; *Brooker et al., 2022*). For example, *Jensen & Morse (1990)* identified an inductive organic molecule that induced larval settlement in marine polychaete *Phragmatopoma californica* in the laboratory and also triggered the same processes in the ocean. Previous research on horseshoe crabs, to the best of our knowledge, has not compared the potential behavioral difference between hatchery-bred individuals and those in the field. However, given that horseshoe crab populations are threatened and even endangered across the distribution range, previous studies suggest the use of artificially cultured horseshoe crabs is useful to explain the habitat selection mechanisms and distribution patterns of the wild populations (*e.g.*, *Medina & Tankersley, 2010*; *Hieb et al., 2015*; *Kwan et al., 2020*; *Chan et al., 2022*). Apart from this, horseshoe crabs are also likely to use multiple sensory cues, particularly visual cues, in settlement habitat selection. As the entire exclusion of the multisensory factors is challenging, in this study, we can observe some discrepancies in the juvenile orientation results. In Fig. 2, the juveniles showed (1) avoidance behavior toward mangrove *A. marine* cues at 2.00 g l$^{-1}$, but not at 0.25, 0.50 and 1.00 g l$^{-1}$; and (2) there is neither preference nor avoidance of cordgrass *S. alterniflora* except at the highest concentration of 2.00 g l$^{-1}$. Therefore, consideration of the simultaneous use of multiple sensory cues is needed in horseshoe crabs to make further conclusions on the process and mechanism of juvenile habitat selections. Another

possibility of the discrepancies is due to the lower resolution of orientation data compared to the other behavioral parameters: multiple video recordings from a group of juveniles were required to obtain each percentage orientation sample value, but only one video recording per juvenile would be needed to collect each of the other behavioral parameter values.

Collectively, our results demonstrated the differential orientation behaviors of juveniles between seagrass and mangrove/cordgrass chemical cues, which may provide useful navigation to juvenile *T. tridentatus* to identify and settle on the upper intertidal flats adjacent to seagrass habitats, and avoid getting too close to densely vegetated areas of mangroves and saltmarsh cordgrass. The results of nursery habitat selection can maximize the chance to obtain high-quality food and avert adverse environmental conditions, and thereby increasing the survival rate of the juveniles. Additional emphasis on mating, food searching and predation avoidance should also be addressed in *T. tridentatus* and other Asian species to make further conclusions on the role of chemical cues in horseshoe crabs. From a management perspective, preserving coastal and estuarine habitats, particularly those with seagrass beds, should be prioritized in management measures for conservation of the declining Asian horseshoe crab populations. Active seagrass restoration in the upper and middle portion of intertidal areas can also benefit Asian horseshoe crab conservation by providing more suitable nursery habitats for shelter and basal production sources in the juvenile food web.

## CONCLUSIONS

Our findings demonstrated that early-stage juvenile *T. tridentatus* are capable of detecting and responding to varying sources of habitat vegetation. Positive rheotaxis was exhibited in the presence of seagrass *H. beccarii* cue at 0.5 and $1 \, \mathrm{g \, l^{-1}}$, but juveniles avoided mangrove *A. marina* and invasive saltmarsh cordgrass *S. alterniflora* cues when the concentrations were too high at $2 \, \mathrm{g \, l^{-1}}$. Juvenile displacement was also significantly reduced in water containing *S. alterniflora* cue at 1 and $2 \, \mathrm{g \, l^{-1}}$. These behaviors may help guide juveniles to high-quality settlement habitats, as seagrass is known to serve as basal production sources in the *T. tridentatus* food web, as well as prevent juveniles from getting too close to the mangrove and saltmarsh cordgrass, which are generally higher in organic matter and hydrogen sulfide. The present study provided valuable evidence on the scope of larval dispersal and habitat selection mediated by habitat chemical cues, which is useful to improve the management efforts for the declining Asian horseshoe crab populations.

## ACKNOWLEDGEMENTS

We would like to thank the students from College of Marine Sciences of Beibu Gulf University for their assistance in conducting field sampling and laboratory maintenance. Constructive comments by the Editor, Dr. Sheri Johnson, as well as Dr. Mark Botton, Dr. Kiran Liversage and the anonymous reviewer are much appreciated.

### Funding

This work was funded by the National Natural Science Foundation of China (32060129), the Beibu Gulf Ocean Development Research Centre under Key Research Base of Humanities and Social Sciences in Guangxi Universities, Marine Science Program for Guangxi First-Class Discipline, Beibu Gulf University (DRA002, TRA001), and the Guangxi Recruitment Program of 100 Global Experts. The funders had no role in study design, data collection and analysis, decision to publish, or preparation of the manuscript.

### Grant Disclosures

The following grant information was disclosed by the authors:
National Natural Science Foundation of China: 32060129.

### Competing Interests

The authors declare there are no competing interests.

### Author Contributions

- Kit Yue Kwan conceived and designed the experiments, performed the experiments, analyzed the data, prepared figures and/or tables, authored or reviewed drafts of the article, and approved the final draft.
- Xin Yang conceived and designed the experiments, performed the experiments, analyzed the data, prepared figures and/or tables, and approved the final draft.
- Chun-Chieh Wang analyzed the data, authored or reviewed drafts of the article, and approved the final draft.
- Yang Kuang conceived and designed the experiments, performed the experiments, prepared figures and/or tables, and approved the final draft.
- Yulong Wen performed the experiments, authored or reviewed drafts of the article, and approved the final draft.
- Kian Ann Tan analyzed the data, authored or reviewed drafts of the article, and approved the final draft.
- Peng Xu analyzed the data, authored or reviewed drafts of the article, and approved the final draft.
- Wenquan Zhen analyzed the data, authored or reviewed drafts of the article, and approved the final draft.
- Xueping Wang conceived and designed the experiments, prepared figures and/or tables, and approved the final draft.
- Junhua Zhu conceived and designed the experiments, authored or reviewed drafts of the article, and approved the final draft.
- Xing Huang conceived and designed the experiments, authored or reviewed drafts of the article, and approved the final draft.

### Data Availability

The raw measurements are available in the Supplementary File.

## Supplemental Information

Supplemental information for this article can be found online at http://dx.doi.org/10.7717/peerj.14465#supplemental-information.

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
