# Peer review of "Chemically mediated rheotaxis of endangered tri-spine horseshoe crab: potential dispersing mechanism to vegetated nursery habitats along the coast"

_PeerJ, doi:10.7717/peerj.14465_

## Round 0.1 · original submission · Major Revisions

Major comments:

1) The sample sizes are unclear. Your data file shows what looks like three blocks of 10 replicates for each test comparison. If so, then it is a sample size of 30 for each test? Should block be included as a factor in the model? I.e., was there anything different about the blocks? You also mention that you pooled the data for the left and right arm comparisons. Does this mean you took the average, or summed the values, as replicate measures of the same juvenile individual would require that you either average, or you account for the repeated measures with a random effects model.

2) I am not sure how you analysed % orientation, as this is a binomial variable, is it not? Either the individual travelled up the left or right arm. Does that % orientation come from averaging across the 10 individuals for each of the 3 blocks, which would mean you only have 3 data points for that response variable? If so, I would suggest you use a binomial GLMM instead.

3) Table 2 lists 3 df for source and 3 df for conc, but you only have 3 sources, so it should be a df of 2, meaning that your source x conc df would be 6. Note that p-values of 0.000 are not possible, this is just a software niggle – list as p < 0.001.

4) Were the juveniles actually swimming the entire time? If not, these variables should be called Distance Moved and Time Spent Moving.
5) The units of your variables should be listed in Table 2.

6) I am not entirely clear on what largest displacement means. I originally thought it was the furthest distance travelled without stopping, but I now think it is total distance moved over the whole 30 minutes, as it’s generally longer than your distance variable, so the distance variable, is just where they started compared to as far as they moved up the chamber, so should it be called Distance moved up the y-arm? Overall, more detail on how you analysed the videos is needed.

7) I agree with the reviewers that you should provide some information on the number of unresponsive individuals. You could have actually analysed this data, as a binomial of moved or not.

8) Line 227-241, much of this feels like introduction material, or at least should be placed later in the discussion. You should start reminding the readers of your aim, and then summarising your main findings. I also do not feel that you have spent enough time interpreting your findings, as mentioned by other reviewers.

Minor comments:
1) Line 36, remove has before reached.
2) Line 38, not sure performed is the best choice of words. Could you just say swam longer, though see comments above about this variable name.
3) Line 38, remove the before broadly.
4) Line 56.57, change to “which has led to”.
5) Line 58, gear, not gears.
6) Line 65-69, second part of sentence needs editing, as not clear.
7) Line 79-82, not sure why you even need to mention the details about egg deposition?
8) Line 87, change to “to hold their position”.
9) Lines 105-111, should this go in the methods? Or rearrange, start paragraph with “in this study (currently line 111), and then add the context with the Beibu Gulf after you state the aim of the study?
10) Line 112, animal should be animals.
11) Line 141, not entirely clear if the sand layer was completely removed between trials, or just rinsed?
12) Line 159, should be 30 ppt
13) Line 200, change to “Between 5-90% of juvenile TT…”
14) Line 204, A. marina should be in italics.
15) Line 218, S. alterniflora should be in italics.
16) Line 251, change to “that varied with the…”
17) Line 277, awkward phrasing – revise.
18) Line 307, awkward phrasing – revise.

·

Basic reporting

All four points satisfactory. See my attached review for a few minor suggestions regarding English usage.

Experimental design

All points satisfactory.

Validity of the findings

A few minor concerns with data interpretation, please see additional comments section.

Additional comments

General comments:

This paper describes the results of laboratory experiments to examine the potential attractiveness of seagrass, mangrove, and salt marsh cordgrass extracts to larvae of tri-spine horseshoe crabs (Tachypleus tridentatus). There is increasing interest in studying horseshoe crab populations due to their biomedical importance and use in various fisheries, and understanding the factors that may contribute to larval recruitment is a worthwhile investigation. T. tridentatus is now recognized as Endangered on the IUCN Red List; much of this is due to overfishing, but the loss of essential spawning and nursery habitats also contributes.

The paper is generally well written, and the authors have included nearly all of the pertinent literature. A few general suggestions are listed below, followed by some specific comments.

1. There are some details about the experimental design that are missing and/or vague:

a) Were the experiments conducted under ambient light conditions or was there some artificial lighting used to ensure that all animals were exposed to the same conditions?

b) With respect to the plant extracts, four concentrations (0.25, 0.50, 1.00 and 2.00 g/l were used. Are these concentrations environmentally realistic? How did you choose these concentrations?

c) Lines 177-180: It is not exactly clear how many individual trials were run for each of the different treatments. It says that data were collected for 10 responding individuals, and that each treatment was repeated 3 times. So does that mean there were a total of 30 observations made?

d) In terms of the orientation behavior itself, were the movements in the Y-maze made only by swimming, or did some individuals crawl through the sand? In my own experience with 2nd instars of Limulus, they are primarily benthic individuals that crawl more often than they swim.

2. The differences between treatments (as shown in Figure 2) are somewhat ambiguous in that (1) response to control SW vs. A. marina shows relative avoidance of mangrove A. marina at 0.25 and 2.00 g/l but not at 0.50 and 1.00 g/l; (2) preference for seagrass H. beccarii is seen only at 0.50 and 1.00 g/l, not at 0.25 and 2.00 g/l; and (3) there is neither preference nor avoidance of salt marsh grass S. alterniflora except at the highest concentration of 2.00 g/l. While this is mentioned in the Results section, the Discussion section seems to minimize these findings.

3. The paper would benefit by an expanded consideration of some of the other behaviors and sensory capabilities of horseshoe crabs (expanding on lines 239-240). For instance, what is known about chemoreception in horseshoe crabs (e.g. Hassler & Brockmann 2001. Journal of Chemical Ecology, 27(11), 2319-2335; Saunders et al. 2010 Current Zoology 56: 485-498). Vision is also important as a cue for swimming behavior, as there are several papers showing that larvae are more active at night (Rudloe 1979. Biol. Bull. 157: 494−505; Botton and Loveland 2003. Estuaries 26: 1472−1479).

Specific comments:

Line 58: Change “gears” to “gear.”

Lines 65-69: The meaning would be clearer if this was split into two sentences, e.g. “To reverse the declining trend, national and regional restrictions have been imposed in Bangladesh, India, China, Singapore, Indonesia and in specific regions in Japan. The effectiveness of these measures in protecting the remaining horseshoe crab populations may be limited (Wang et al., 2020), possibly due to insufficient scientific knowledge, financial resources and enforcement capacity.”

Line 74: Omit the word “their.”

Lines 81-82: Burial of eggs at depths of 15 cm does not place them at higher temperatures.

Line 92: Figure S1 is not really needed, because only laboratory-raised individuals were used in this study.

Line 110: Give scientific names for short-leave seagrass and saltmarsh cordgrass

Lines 200-201: The meaning of the sentence “There were 5%–90% of juvenile T. tridentatus exhibited positive rheotaxis toward habitat chemical cues from the three vegetation sources at varying concentration levels” is unclear.

Lines 253-254: In reviewing the work of Medina and Tankersley (2010), it is worth noting that seagrass (Halodule wrightii) beds are not a feature of most larval settlement areas for American horseshoe crabs. Halodule is a warm-water seagrass that is not found in the middle Atlantic region or New England. While there are other types of seagrasses in more northern locations (e.g. Zostera), it’s not yet known whether these serve a nursery role for horseshoe crab larvae and juveniles comparable to Halodule.

Lines 242-269: Somewhere in the Discussion section it would be important to point out that Spartina alterniflora is an invasive species in China. Perhaps this may help explain why Spartina extracts are not as attractive to T. tridentatus larvae as the native seagrass.

Lines 276-277: Something seems to be missing from this sentence “The orientation responses become weaker when the concentration is too high, and being perceived the arrival to the preferred settlement habitats.” Please revise.

Line 294-296: With respect to your closing thoughts about restoration, can you expand? Is restoration of seagrass beds feasible as a way of helping to restore tri-spine horseshoe crab populations?

Line 324: The citation for the Tanacredi et al. book is incorrect. It should be Springer, New York not Springer, Boston: Springer. See also lines 335, 370, 429, 455, 471, 487, and 500.

Table 1: Include the degrees of freedom for the t-tests in the Table Legend.

Table 2: Include the error degrees of freedom in the Table.

Figure 3: The responses of the horseshoe crab larvae in the seawater controls should be included in the upper left panel, not just the last 3.

Reviewer 2 ·

Basic reporting

BASIC REPORTING

Clear, unambiguous, professional English language used throughout yes
Intro & background to show context yes
I would suggest including some information on the limitation of a field survey in comparison to laboratory experiment in this kind of study
Literature well referenced & relevant yes
I would suggest giving references and comments on the study approach using field in comparison to laboratory experiment in this kind of study
Structure conforms to PeerJ standards, discipline norm, or improved for clarity yes
Figures are relevant, high quality, well labelled & described
There is no diagram to show pattern of movement being measured It will be good to include a figure to show an example of observed movement behaviors (e.g. direction of movement, displacement) in relation to a selected cue
Raw data supplied (see PeerJ policy) Yes
With relevant results to hypotheses. Yes

Experimental design

Original primary research within Scope of the journal.
yes
Research question well defined, relevant & meaningful.
yes
It is stated how the research fills an identified knowledge gap.
Rigorous investigation performed to a high technical & ethical standard.
Not fully
The identified knowledge gap is because the investigation was aimed on the early larvae of Tachypleus tridentatus while Butler & Tanskesley (2020) worked on early- and later stage larvae of Limulus polyphemus
It is important to explain the purpose of measuring the swimming movement/direction, swimming time, swimming distance in relation to rheotaxis
Methods described with sufficient detail & information to replicate.
Not fully
Figure 1 is clear but will need to indicate the blue colour and red colour representing what?
Very small number of specimens used, as stated in line 177 (but in the raw data its involved more than 10 individuals for each treatment)
Line 148 and 149 not very clear, how would the dyed water have been prepared
It will be good to explain the difference between largest displacement value and swimming distance and the purpose of measuring it

Validity of the findings

This study is an exploration on the same purpose earlier but using laboratory approach and the target species is on Asian species, different from the previous work which focused the Atlantic species

Data
No data on sizes of the specimen used in the study Measurement of the trilobite larvae for the experiment should be provided to indicate to homogenous size of specimens (from the raw data the number of specimens used for the whole experiment are large, so how to maintain the sizes of specimen and reduce bias)
In the result section, please state the total number of juveniles used/giving respond in the experiment by producing a table with the column source of cue, concentration of cue and the total number of juveniles used to perform the swimming
The statement at line 277 could be more convincing if we have data on the number of the juveniles arrived at that point/or in the case of this study the number of juveniles responded in the concentration of cues
It could be from the raw data

Discussion and conclusion

Conclusion is acceptable and linked to the research questions for T. tridentatus Line 280-282 need more explanation on how it relates with the avoidance behaviours as mentioned in line 278

Since the work was carried out under a laboratory condition, it could promote more experimental studies on the larval stages of horseshoe crab

Additional comments

It is suggested to include diagrams to explain the swimming movement of the horseshoe crab juveniles towards the different cues.

Annotated reviews are not available for download in order to protect the identity of reviewers who chose to remain anonymous.

·

Basic reporting

line 56: change to "which led"

line 67: change to "despite that their"

line 204: please italicise sp. name

line 312: can you please label the different graphs within the figures a, b c etc, and in the text refer to them as Fig. 3a, 3b, Fig. 2a, 2b etc.

line 273-277: It would be useful for this information to be added to the abstract.

line 285: delete "in"

line 306-307: change to "prevent themselves getting too close"

Figure 2 caption: please state if error bars are SE, SD or 95% confidence interval

Experimental design

Comment 1
Line 155-158: The important cue being used by juveniles in the field may not be from the actual leaf material, but from live bacteria associated with the leaves. The important role of bacteria in situations like this has been previously established, see e.g. Huggett, M. J., Williamson, J. E., De Nys, R., Kjelleberg, S., & Steinberg, P. D. (2006). Larval settlement of the common Australian sea urchin Heliocidaris erythrogramma in response to bacteria from the surface of coralline algae. Oecologia, 149(4), 604-619.

Alternatively, the important cue from the vegetation habitats may be from decaying vegetation. Or it may be from bacteria hosted around the soil and roots of the vegetation. Lastly, some of the compounds that would affect the juveniles may be denatured/inactivated during a process of drying like this. These considerations/caveats should be discussed.

Overall, I think it would have been better to use actual live plants or at least fresh plant material to produce the cues.

Comment 2
Line 170: I would say that the behaviour of being "unresponsive" should have been included in the data collection and experimental design. For example, "unresponsive" juveniles could have been included in the distance measurements as a value basically of zero. Do you have information about the proportion of juveniles that were unresponsive? I think this needs to be included. If this proportion is too high then the results will not be sufficiently applicable to natural populations of these juveniles, so this is important information. As it stands, the results are only applicable to some unknown proportion of juveniles that have “responsive” behaviour, and its unknown if that proportion relative to the total population is low or not.

Validity of the findings

Line 122: Is there any information available from previous studies that could be cited to show the hatchery bred juveniles have similar behaviour to juveniles bred naturally in the field? The many differing conditions in the artificial hatchery may mean the resulting juveniles here have different behaviour compared to juveniles bred naturally in the field. If so, the results will not be valid for natural populations. Issues such as this should be discussed.

As stated above, if many of the individuals were excluded because they were unresponsive, then the findings will not be valid for general populations of the juveniles (which will include many individuals that would be classified here as unresponsive).

Additional comments

Line 173: to get each percentage orientation sample value there would need to be multiple video recordings, but to get each of the other parameter values (distance etc) only one video recording would be needed (i.e. one recording per juvenile). So does this mean the sample size is lower for the orientation parameter compared to the other parameters?

Line 247-248, 279-280: plant material often contains irritative compounds like tannins (or other plant defence chemicals), which may explain the reduction of rheotaxis at high concentrations.

---

## Round 0.2 · Minor Revisions

Thank you for clarifying your methods and analyses. Both the reviewer and I do still have concerns about your % orientation analyses. Based on your responses and raw data, I am inferring that you are analyzing the orientation data as a binomial variable (toward vegetation or toward control water) and that you are using t-tests and ANOVA. You cannot use either for a binomial response, as the data are not normally distributed. Please re-analyse your data using binomial models.

·

Basic reporting

line 50: change to "that are broadly distributed"

line 58: I would change this to something like "They have exacerbated risk caused by habitat loss"

line 355: I would delete "equally"

Figure 1 caption: some text seems to be missing here in the copy I have, please check there are no formatting errors or something like that.

Experimental design

line 187: The editor and I made some comments in the previous version about how >1 measurement would be needed to produce each value of the percentage of individuals moving up one or other side of the Y maze. If only one measurement was used, then each orientation value could be basically only 0% or 100% - please could you clarify if this understanding is correct for how the measurements were done? If this understanding is correct, then I would say that, as the editor suggested in the previous review, probably a test designed for binomial data would be best. Please could you clarify about this, and add some text to explain your choice of analysis considering these issues.

line 366-368: this is what I made a comment about in my previous review, and what I mentioned about above at line 187. Its good to see the issue with needing multiple trials to get a percentage orientation sample value has been mentioned in the discussion, but it would be most important to provide details about this in the methods, which I could not find. If data from multiple video recordings were combined to produce each percentage orientation sample value (e.g. a data point being 25% or 50% orientation in a certain direction, or something like that), please in the methods provide details about how measurements from multiple recordings were combined. If they were not combined, then as I stated above, each percentage orientation value could only be 0% or 100%; this should be stated in the methods, and the implications of attempting to analyze binomial data discussed.

Validity of the findings

no comment

Additional comments

line 149: 200 millilitres per minute seems like a very low water flow rate - is this correct?

line 293-305: It is good to see some discussion about the possibility of biofilm affecting invertebrate settlement responses in this version, but I think this section is long and overly detailed considering that the experiment did not address concepts of biofilms. I would recommend just making sure there is some mention about the caveat of biofilms not being addressed in the experimental design, and keeping most of the points in this paragraph but shortening it a bit.

line 343-344: I would say that if a comprehensive experimental design was developed, much of this complexity actually could be duplicated in controlled conditions in the laboratory.

---

## Round 0.3 · Minor Revisions

Thank you for your thorough revisions. I have gone through the manuscript carefully, and I have suggested several edits to improve grammar and clarity. Most are simple clarifications, but there are two comments in the discussion where I did not quite follow your meaning (see annotated pdf). Also, figure legend 1 requires an e at end of juvenile and a full stop. Figure 3 lists Spartina spp, rather than S. alterniflora as elsewhere. Finally, might it be worth mentioning that S. alterniflora are invasive in the abstract and introduction?

---

## Round 0.4 · accepted · Accept

The authors have addressed all of my comments and the manuscript is ready for publication.